# Futureproofing ML
## 15+ Dialects, One Compiler (PyTorch, JAX, RDNA, SASS, …)

## Abstract

As machine learning frameworks fragment and hardware architectures evolve, research code rapidly degrades into obsolete syntax. To guarantee **Futureproofing**, we introduce a methodology based on **Sovereign Specifications**, where the model's logic is elevated above its implementation language. We implement this via two extensive compilers: `redacted_project_name0`, a semantic engine supporting **15 source/targets** that spans the entire abstraction stack—translating between high-level frameworks like PyTorch/JAX/TensorFlow and low-level hardware instructions like NVIDIA SASS—and `redacted_project_name1`, which unifies **12 structural interfaces** via AST manipulation. By reducing the $O(N^2)$ complexity of pairwise translation to a linear $O(N)$ hub-and-spoke model, we prove that mathematical intent can be preserved as an immutable artifact, capable of being re-synthesized into whichever runtime environment dominates the future.

## 1. Introduction: The Methodology of "Done"

The history of ML algorithms is as old as mathematics itself, yet the last decade has seen these algorithms locked into increasingly brittle software substrates. Research output has increased at a double-exponential rate (Krenn et al., 2023), but the artifact of this research—Source Code—is paradoxically the weakest link. Publishing code tied to a specific version of a framework (e.g., "PyTorch 1.13") is a "lossy compression" of mathematical intent. It binds the abstract logic to the transient memory management, random number generation, and dispatch mechanisms of a specific runtime which will inevitably deprecate.

When a researcher publishes a model, they are implicitly stating: "This logic is valid within the constraints of Python 3.10 and PyTorch 2.1." Five years later, this statement becomes false. Dependencies break, CUDA drivers shift, and the model dies.

To achieve **Futureproofing**, we define a methodology centered on **Declarative Sovereignty**. A model is not "done" when it runs; it is done when its logic is extracted into a platform-agnostic **Operation Definition Language (ODL)**.

We validate this methodology through a Python reference stack. While implemented in Python for immediate utility, the architecture is fundamentally substrate-independent. We demonstrate:

1. **Ghost Mode**: The effective decoupling of the compiler from the runtime environment. Our implementation, `redacted_project_name0`, analyzes and manipulates framework logic via static snapshots without that framework being executed or even installed.

2. **The API structure Bridge**: How structural compilation (`redacted_project_name1`) enforces regularity that enables robust static analysis (via **Griffe**, at the core of the secondmost popular Python docs generator `mkdocs` (Christie, 2014)), creating a verified input stream for the semantic compiler. A fork of Griffe was developed with a core of `redacted_project_name1`.

3. **Digital Archaeology**: We prove agnosticism by applying the same ODL approach to lift assembly languages of NVIDIA SASS and AMD RDNA back into high-level specs, effectively rescuing logic from compiled silence.

## 2. The Ephemerality of Implementation

To understand the necessity of futureproofing, we must examine the trajectory of the ML substrate. This history reveals a trend: semantics persist, but syntax is ephemeral. An early ML framework was 1994's WEKA (McQueen et al., 1994); originally in C, Makefile, and Tcl/Tk. Four years later it was rewritten in Java (Frank et al., 1999); and

[1] Anonymous Institution, Anonymous City, Anonymous Region, Anonymous Country. Correspondence to: Anonymous Author <anon.email@domain.com>.

Preliminary work. Under review by the International Conference on Machine Learning (ICML). Do not distribute.

is still maintained to this day with GPU accelerated variants interfacing Java with Python (Cassales et al., 2025).

### 2.1. The Configuration Era (Caffe, ProtoBuf)

Early deep learning frameworks like 2013's Caffe (C++ with a Python interface) and later [Caffe 2] rewrites (Dipert, 2016; Team, 2017) defined models not as code, but as data (Protocol Buffers). This decoupled the model structure from the execution engine; an early form of futureproofing and substrate independence. However, the lack of imperative control flow limited research into dynamic architectures and made it more difficult for juniors to approach (Yu et al., 2018).

### 2.2. The Symbolic & Imperative Eras

The field moved to Python proper (Raschka et al., 2020) (with Theano (The Theano Development Team et al., 2016) and TensorFlow (Martín Abadi et al., 2015)) for flexibility. Later, PyTorch revolutionized the field by merging definition and execution ("Eager Mode") (Paszke et al., 2017). While maximizing research velocity, this coupled mathematics tightly to the CPython interpreter and Object-Oriented state management, accumulating technical debt in the form of implicit state (Global RNG, In-place mutation). This debt compounds the complexity of migration to compiled backends (e.g., XLA or Apache TVM) (Li et al., 2021).

### 2.3. The Function Era

Google's part of the ML ecosystem has explicitly rejected the imperative eagerness of the more popular PyTorch with JAX (Frostig et al., 2019)+XLA (JAX: 10M/month vs PyTorch: 61M/month(PyPi, 2026; Tor, 2026)); demanding functional purity. This shift from stateful objects (OOP) to pure functions necessitates a fundamental rewrite of variable handling and RNG.

### 2.4. The [Time-Travelling] Sovereign Era

We propose the **Sovereign Era**, where the **ODL** is the source of truth. By treating Python/C++ code as a mutable projection of the ODL, we enable transparent migration betwixt all eras.

## 3. The Code Rot Paradox

Software engineering typically assumes that code "rots" linearly over time as dependencies change (Le et al., 2018; Li et al., 2022). In Machine Learning, however, we observe a "Code Rot Paradox": the utility of ML source code decays at a significantly accelerated rate compared to traditional systems software, despite the underlying mathematics (e.g., matrix multiplication, backpropagation) remaining constant (Jahan et al., 2025; Latendresse et al., 2025).

This acceleration is driven by the **Hardware/Software Co-design Feedback Loop**. Frameworks like PyTorch and TensorFlow are not merely libraries; they are dynamic interfaces to specific generations of hardware accelerators (GPUs, TPUs). As hardware architectures shift—from NVIDIA Volta to Hopper, or from GPU SIMT to TPU Systolic Arrays—the optimal expression of logic changes, rendering previous syntactic abstractions obsolete or inefficient (Coyne, 2025; Barnes, 2025).

### 3.1. The Tower of Abstractions

We quantify this fragility by modeling the ML stack as a vertical dependency tower. When a researcher writes `y = x @ w`, they are implicitly invoking a dependency chain that spans six layers of abstraction:

1. **High-Level Intent**: The linear algebra ($y = xW$).

2. **Framework API**: The syntax (`torch.matmul`).

3. **Binding Layer**: The C++/Python bridge (pybind11).

4. **Kernel Runtime**: The dispatcher (cuBLAS / CUTLASS).

5. **Instruction Set**: The assembly (SASS / PTX).

6. **Physics**: The transistor logic (Silicon).

Current source code practices are **Vertically Integrated**. A script written in PyTorch 1.0 is tightly coupled from Layer 2 down to Layer 4 of that specific era (Wang et al., 2023). When Layer 4 (kernels) changes to support new hardware, Layer 2 (syntax) often shifts to expose these new capabilities (e.g., the introduction of `channels_last` [NHWC] memory formats), breaking the code.

### 3.2. The ODL Horizontal Cut

The proposed ODL methodology implements a **Horizontal Cut** at Layer 1. By defining the model in a declarative schema, we sever the dependency on the turbulent layers below. The schema defines the mathematical contract; the `redacted_project_name0` compiler is responsible for re-binding that contract to the appropriate vertical stack at compile time, effectively insulating the research artifact from the rapid decay of the hardware/software substrate.

## 4. The Methodology of Sovereign Specifications

Before distinguishing the tools, we delineate the three theoretical pillars of our approach to Futureproofing.

**1. Representation Independence**: Mathematical logic must be stored in a format that does not require a Turing-complete interpreter to parse. If logic requires executing Python to understand what it does (dynamic graphs), it is not futureproof. Our ODL uses declarative YAML to define constraints, types, and logic, ensuring the "Map" survives the "Territory."

**2. Structural Regularity**: Interfaces must be consistent. A model definition, a CLI argument parser, and a database schema should all be projections of a single Truth, not manually synchronized copies. This creates a surface area that automated tools can analyze reliably without executing code.

**3. Semantic Sovereignty**: The abstract definition of an operation (e.g., `Conv2d`) is sovereign over its implementation. If PyTorch's `Conv2d` differs from the mathematical standard, the implementation is treated as an "Adapter" to the standard, not the standard itself.

This methodology dictates that we need two distinct tools: a **Structural Compiler** to enforce regularity (Pillar 2), and a **Semantic Compiler** to manage sovereign definitions (Pillar 3).

## 5. The Substrate Independence Thesis

True futureproofing requires independence not just from software syntax (e.g., Python vs. Mojo vs. Rust), but from the fundamental execution semantics of the underlying hardware buffers. The current ecosystem is bifurcated by two distinct execution paradigms: **Eager Execution** and **Staged Compilation**, creating a synchronization gap that source-to-source translation must bridge.

### 5.1. Eager vs. Staged Semantics

Frameworks like PyTorch primarily utilize **Eager Execution**: calls to the API (e.g., `torch.add`) dispatch kernels immediately to the GPU. The program counter blocks or the runtime syncs implicitly, ensuring that the result is conceptually available on the next line of Python.

Conversely, frameworks like JAX and TensorFlow utilize **Lazy** or **Staged Compilation** (via XLA). An API call does not execute math; it emits a graph node into a tracer. Execution is asynchronous, and the result is a "Promise" (future) rather than a value.

### 5.2. Normalization of Time

The ODL acts as a normalizing buffer for these divergent timelines. ODL definitions allow for the specification of **Execution Constraints**.

For example, when lifting imperative PyTorch code that relies on immediate side-effects (e.g., printing a tensor value

for debugging) to JAX, a naïve translation would fail because the JAX tracer would print the abstract Tracer object, not the computed value.

The `redacted_project_name0` engine handles this via the `AuxiliaryPass`. If the ODL specifies strict timing requirements, or if the `PurityScanner` detects IO dependency, the compiler injects explicit synchronization barriers (e.g., `jax.block_until_ready()`) or wraps the operations in a callback (e.g., `jax.debug.print`). This proves that the Logical Graph ($G$) defined by the ODL is sovereign: it imposes its own temporal requirements onto the target substrate, rather than succumbing to the default behavior of the runtime.

## 6. System I: The Semantic Knowledge Base

The intelligence of our reference implementation, `redacted_project_name0`, is not hardcoded in the compiler logic, but stored in the **Semantics Manager** (The Hub). This distributed database enables the separation of the *Specification* from the *Implementation*.

### 6.1. Hub-and-Spoke Architecture

Direct $N$-to-$N$ translation requires $N^2 - N$ converters. We employ a strict Hub-and-Spoke model where every framework maps to a central "Abstract Standard". This reduces complexity to $2N$.

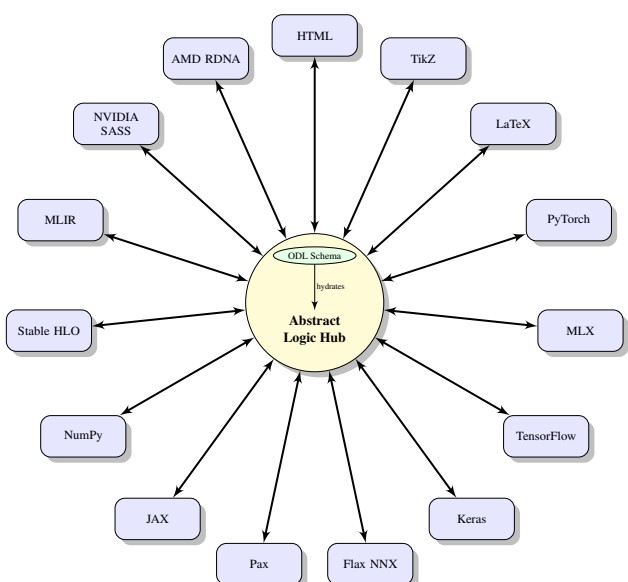

*Figure 1.* The Hub-and-Spoke Architecture. The ODL Schema (center) hydrates the Abstract Logic Hub, ensuring that logic is represented as data, not code, enabling projections to 15+ distinct representations. New target/source languages/frameworks can be added by adding a few new Python & JSON files to the Python module hierarchy (no existing files need altering).

- **The Hub (Specs):** JSON definitions of abstract operations (e.g., `LogSoftmax`) based on `src/redacted_project_name0/semantics/*.json`. These define *what* the operation implies mathematically.

- **The Spokes (Snapshots):** Overlay files mapping specific frameworks (e.g., `torch v2.1`) to the Hub. These define *how* to invoke the operation.

### 6.2. Operation Definition Language (ODL)

The ODL schema (defined in `src/redacted_project_name0/core/dsl.py`) uses Pydantic to enforce rigor. Unlike simple name mapping, ODL captures constraints required for fuzzing and correctness verification.

*Listing 1.* ODL Definition for LogSoftmax

### 6.3. Ghost Mode: Agnostic Introspection

A key validation of our agnostic methodology is **Ghost Mode**. Traditional introspection requires installing a library to analyze it via Python's builtin `inspect` module. This is brittle and platform-dependent; e.g., one cannot inspect PyTorch from a WebAssembly browser environment.

Ghost Mode utilizes **Snapshots**; serialized dumps of the library's API surface stored as JSON. The compiler loads these snapshots via `src/redacted_project_name0/core/ghost.py` to validate code against a framework definition *without* the framework being present. This enables our Python implementation to run in restricted environments like WebAssembly, proving that the intelligence is in the Data, not the Runtime.

## 7. System I Execution: The Engine

The `redacted_project_name0` runtime engine (`src/redacted_project_name0core/engine.py`) implements a Dual-Path architecture, proving that the methodology can handle both structured high-level code and flat low-level streams.

### 7.1. Path A: The Rewriter Pipeline (High-Level)

For Python-to-Python transcoding (e.g., Torch to JAX), we use a Concrete Syntax Tree (Meta's Instagram's LibCST) approach to preserve comments and formatting.

#### 7.1.1. CONCEPT LINKING VIA STATIC STRUCTURAL ANALYSIS

A robust methodology requires analyzing source code logic without executing it. To achieve this in our Python reference implementation, we leverage **Griffe**

---

**Algorithm 1** The Semantic Pivot Strategy

**Input:** AST Node $N$, Source $F_S$, Target $F_T$, Hub $H$.
**if** $N$ matches API in $F_S$ **then**
    $Op \leftarrow \text{LookupAbstractOp}(N, F_S, H)$
    $Args \leftarrow \text{NormalizeArgs}(N, Op)$
    $Traits \leftarrow \text{GetTraits}(F_T)$
    *# Phase 1: Logic Injection*
    **if** $Traits.requires\_explicit\_rng$ **then**
        $Args \leftarrow \text{InjectRngKey}(Args)$
    **end if**
    *# Phase 2: Projection*
    $TargetImpl \leftarrow \text{LookupImplementation}(Op, F_T)$
    $N' \leftarrow \text{GenerateCall}(TargetImpl, Args)$
    **return** $N'$
**end if**
**return** $N$

---

for static analysis. However, dynamic Python code is often resistant to static analysis. This is where **System II** (`redacted_project_name1`) becomes critical. `redacted_project_name1` enforces structural regularity, ensuring that APIs are "lifted" into a canonical form that Griffe can reliably parse. And as referenced, a Griffe-fork maintained by *The Authors* swaps out its internal implementation for `redacted_project_name1` also.

This creates a verified pipeline: Unstructured Code → `redacted_project_name1` → Structured APIs → Griffe Analysis → `redacted_project_name0` → ODL match.

#### 7.1.2. PURITY ANALYSIS: THE SAFETY VERIFICATION

Futureproofing requires identifying code that relies on specific runtime side-effects (e.g., global RNG). The `PurityScanner` detects I/O, global mutations, and hidden state, warning the user that the code relies on a specific substrate rather than pure logic. This is critical for targeting XLA-based frameworks like JAX, where side effects in compilation traces can cause silent failures.

#### 7.1.3. PLUGIN ARCHITECTURE: BRIDGING PARADIGMS

Semantic gaps (OOP vs Functional) are bridged via hooks. A prime example is **RNG Threading**. The `rng_threading` plugin transforms implicit global state (PyTorch `manual_seed`) into explicit key passing (JAX `random.PRNGKey`) by injecting arguments into function signatures and splitting logic into the AST dynamically.

*Listing 2.* RNG Threading Implementation in Python

## 7.2. The Import Resolution Graph

A naïve rewriter that converts `torch.nn.Linear` to `flax.nnx.Linear` without managing imports will produce broken code. The redacted_project_name0 engine addresses this via the **Import Fixer** (`src/core/import_fixer`), which treats dependency resolution as a graph problem.

The `ImportResolver` operates in three phases:

- **1. Usage Scanning:** The AST is scanned to build a symbol usage map. This identifies which source frameworks are actually active (e.g., is `torch` referenced?) and which target aliases are required.

- **2. Resolution Planning:** The Semantic Knowledge Base provides an "Import Map" for the target framework. The resolver generates a `ResolutionPlan` that dictates:
  - **Injection:** Adding `import jax.numpy as jnp` if any math op was converted to JAX.
  - **Pruning:** Marking `import torch` for removal if all its usages were successfully rewritten.

- **3. Tree Shaking:** The `ImportFixer` transformer executes the plan. It performs "Tree Shaking" to remove unused imports, ensuring the transpiled file has zero dependencies on the original framework, fulfilling the requirement of a standalone artifact.

### 7.3. Graph Optimization and Fusion

While 1:1 API mapping ensures functional correctness, it often results in suboptimal code. A sequence like `Conv2d` $\rightarrow$ `BatchNorm` $\rightarrow$ `ReLU` is idiomatic in PyTorch but is often represented as a single fused block in optimized inference kernels or intermediate representations (like StableHLO).

To bridge this gap, the `GraphOptimizer` pass implements a **Greedy Pattern Matching** algorithm on the Logical Graph before code synthesis. This allows the compiler to recognize high-level intent from low-level atomic operations.

This mechanism allows redacted_project_name0 to not just translate syntax, but to upgrade architecture definitions to use modern, optimized primitives available in the target framework.

## 8. System II: The Structural Compiler

The second barrier to futureproofing is boilerplate "glue code." **redacted_project_name1** (Structural Compiler) lifts this surface into a **Structural IR**, decoupling *Interface* from *Implementation*.

---

**Algorithm 2** Greedy Graph Fusion

**Input:** Logical Graph $G$, Patterns $P$ (e.g. CBR)
$Nodes \leftarrow$ TopologicalSort($G$)
**for all** $n \in Nodes$ **do**
   $Sequence \leftarrow [n]$
   **for all** $p \in P$ **do**
      **if** $n.kind == p.head$ **then**
         $Match \leftarrow$ TraceForward($n, p.sequence$)
      **end if**
   **end for**
**end for**
**return** $G$

---

### 8.1. The Structural-Semantic Link

redacted_project_name1 is not just a utility; it validates the redacted_project_name0 methodology. By transforming messy Python classes into clean Intermediate Representations, it proves that the interface definition can be stored as data, allowing [bidirectional] generation/synchronization of `classes`, `functions`, `argparse` functions, `SQLAlchemy` models, and `Pydantic` schemas.

#### 8.1.1. THE "ZERO-COST" ABSTRACTION

Conventional schema tools like Pydantic operate at runtime: they require the python interpreter to load the module, execute the class definition, and instantiate objects to validate structure. redacted_project_name1 operates as a **Zero-Cost Abstraction** because it parses the Python AST directly from the source file text, *without execution.*

This distinction is critical for Futureproofing. A model definition file from 2018 may fail to import today due to missing dependencies (e.g., an old version of 'numpy'). However, 'redacted_project_name1' can still parse that file's AST, extract the schema, and lift it into a modern representation. By treating the class definition as a static schema document rather than executable code, we create a "Universal Interface Definition" that survives the death of the runtime environment.

## 9. System III: Digital Archaeology (Path B)

To prove the language-agnostic nature of our methodology, we include a backend for **Hardware Assembly**. redacted_project_name0 implements a **Graph Compiler** (`core/compiler`) that operates outside the Python Rewriter Pipeline. This "lifts" low-level NVIDIA SASS and AMD RDNA assembly back into high-level logical graphs using topological analysis.

**Algorithm 3** Structural Projection (`redacted_project_name1`)

---

**Input:** Source $S$, Targets $T = \{\text{SQL}, \text{Run}, \text{JSON}\}$
$IR \leftarrow \text{Parse}(S)$ (Extract via AST & Docstring)
**for all** $t \in T$ **do**
    $Emitter \leftarrow \text{GetEmitter}(t)$
    $Schema \leftarrow Emitter(IR)$
    **if** $t$ is SQL **then**
        Generate DDL (SQLAlchemy)
    **else if** $t$ is JSON **then**
        Generate Validation Schema
    **end if**
**end for**

---

### 9.1. Heuristic Lifting

The `SassLifter` and `RdnaLifter` work inversely to standard compilation. They use heuristics to identify control flow structures that imply high-level semantic intent.

Specifically, the `SassAnalyzer` scans instruction blocks for the `ISETP.LT` (Integer Set Predicate Less Than) instruction. In NVIDIA SASS, this instruction is the standard idiom for loop control. By detecting `ISETP.LT P0, R2, 3, PT`, the analyzer infers a loop bound of 3. If this loop encloses memory loads and FMAs (Fused Multiply-Adds), we deduce a convolution operation with `kernel_size=3`. This metadata promotes a raw sequence of ALU ops into a semantic `Conv2d` node in the Logical Graph.

### 9.2. Symbolic Register Allocation

To synthesize valid assembly from the abstract ODL, `redacted_project_name0` implements a **Symbolic Register Allocator** (`backends/sass/synthesizer.py`).

The ODL graph operates on symbolic variable names like `input_tensor` or `weight`. The hardware operates on physical registers like `R0` through `R255`. The `RegisterAllocator` maintains a "Liveness Map." When synthesizing code, it:

1. Assigns a physical register (e.g., `R4`) to a symbolic variable (`input`) upon first use.

2. Tracks the lifecycle of that variable through the instruction stream.

3. Frees the register back to the pool once the variable is no longer referenced in the graph.

This allows us to re-compile 2015-era logic (e.g., Kepler SASS) onto 2025-era hardware (e.g., Blackwell) by simply swapping the backend emitter, while the logical intent remains sovereign in the ODL.

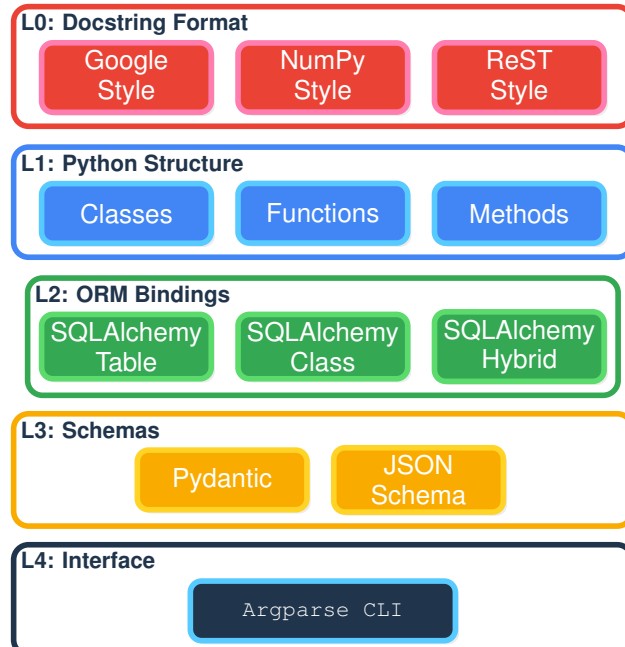

*Figure 2.* Supported parsing and emission types in the Structural Compiler `redacted_project_name1`. This supports a full $O(N^2)$ conversion between all types. And additionally betwixt types in annotation and in docstring.

*Listing 3.* Raw SASS Kernel

*Listing 4.* Lifted Logic

*Figure 3.* Decompilation of an NVIDIA SASS Kernel. The Graph Compiler reconstructs logic from raw assembly.

## 10. System IV: Living Documentation

In the context of futureproofing, documentation must be compiled directly from source code to avoid drift. we provide backends (`backends/extras.py`) that target visualization languages.

### 10.1. Visual Backends (LaTeX, TikZ, & HTML)

The **TikZ Backend** projects the `LogicalGraph` into LaTeX code, allowing researchers to generate publication-quality figures directly from the Python implementation. This ensures the diagram is always synchronized with the code. Similarly, the **HTML Backend** generates interactive Grid layouts for web-based introspection, using the same ODL source of truth. With sufficient *chutzpah* one can now write/maintain their ML models in HTML | LaTeX | TikZ.

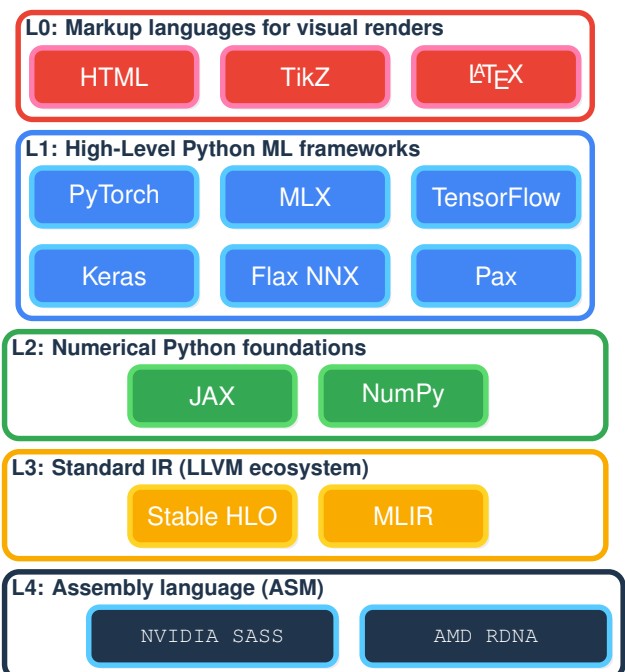

*Figure 4.* Supported frameworks and compilation stack of `redacted_project_name0`; ranging from Visual Markup (L0) down to Assembly (L4).

### 10.2. The WASM "Ghost" Demo

To prove the resilience of the architecture against the "substrate crisis," we compiled `redacted_project_name0` to WebAssembly (Pyodide). Because **Ghost Mode** relies on JSON Snapshots rather than importing heavy libraries, the compiler runs client-side in a browser, despite the absence of source libraries (PyTorch), demonstrating a scenario where the original runtime is unavailable but the logic remains accessible.

## 11. System V: Automated Discovery (The Cyborg Workflow)

Specifications are never "finished"; frameworks evolve daily. To prevent the ODL from becoming a stale artifact, we implement a "Cyborg Workflow" that combines robotic scanning with human oversight to continuously update the Knowledge Base. This is implemented via three CLI tools in the reference stack:

- `scaffold`: Heuristically scans a new library to generate a skeleton mapping.

- `suggest`: Generates an LLM prompt containing introspection data for a specific API, allowing an AI agent to write the ODL.

- `harvest`: Inspects manual test files written by developers. If a developer writes a manual test to fix a bug, the Harvester "learns" the correct mapping from that code and updates the ODL automatically.

### 11.1. The Consensus Algorithm

The heart of this system is the **Consensus Engine** (`discovery/consensus.py`). It solves the problem of vocabulary mismatch (e.g., PyTorch calls it `erf`, JAX calls it `scipy.special.erf`).

The algorithm operates in three steps:

1. **Ingestion**: It scans the API surface of all installed frameworks (the Spokes).

2. **Normalization**: It strips framework-specific prefixes/suffixes (e.g. `torch.`, `.functional`, `Loss`).

3. **Clustering**: It computes the **Levenshtein Distance** between all normalized tokens. Tokens that cluster within a similarity threshold (0.8) are proposed as a "Candidate Standard."

For example, `HuberLoss` (Torch) and `huber_loss` (JAX) normalize to `huber` and cluster together. The engine then proposes `Huber` as the Sovereign Name in the ODL.

## 12. Verification & Self-Healing Specs

A specification is only valuable if it is correct. We implement a rigorous verification loop using Property-Based Fuzzing (`Hypothesis`). The `SemanticsBisector` implements "Self-Healing": if a test fails, it incrementally relaxes constraint tolerances (e.g. from 1e-5 to 1e-3) or updates the Spec JSON to match reality, preventing "spec rot."

## 13. Comparison with MLIR

The primary alternative for ML interoperability is **MLIR** (Multi-Level Intermediate Representation). While MLIR is powerful, it is designed for *Compiler Engineers*, typically operating on lowered traces (linear algebra) rather than source abstractions.

## 14. Evaluation

We choose YAML for ODL to maximize **Human-Readability** and **Fuzzability**. A researcher can manually patch a YAML spec in a text editor; patching an SSA graph in MLIR requires specialized toolchains, and one is not expected to write entire models in MLIR; instead one is meant to use a high level framework that will be passed into a compiler (e.g., XLA, e.g., with StableHLO in the middle) that then produces the MLIR.

*Figure 5.* Cross-framework `redacted_project_name0` translation progress. The table enumerates the number of API endpoints (e.g., `conv2d`, `einsum`) successfully mapped from the Source (rows) to the Targets (columns) via the Semantic Hub. NOTE: A preceding baseline of 179 operations from Array API Standard (2024.12) and 198 operations from ONNX Operators (v1.20.0) are also added.

| Source API | Out of | Target Support (Count) | | | | | | | |
|---|---|---|---|---|---|---|---|---|---|
| | | jax | nnx | pax | keras | mlx | torch | tf | numpy |
| jax .numpy.* | 354 | 354 | 354 | 354 | 123 | 107 | 219 | 121 | 339 |
| flax .nnx.* | 204 | 88 | 204 | 77 | 104 | 89 | 147 | 89 | 4 |
| pax (praxis) | 119 | 3 | 27 | 119 | 31 | 23 | 47 | 13 | – |
| keras .{layers,losses,optimizers}.* | 229 | 28 | 88 | 49 | 229 | 70 | 147 | 203 | 1 |
| mlx .{core,nn}.* | 326 | 226 | 160 | 90 | 184 | 326 | 271 | 179 | 161 |
| torch .* | 951 | 506 | 307 | 248 | 357 | 304 | 951 | 427 | 416 |
| tf .{math,numpy}.* | 300 | 260 | 241 | 241 | 201 | 177 | 256 | 300 | 241 |
| numpy .* | 472 | 434 | 320 | 320 | 170 | 176 | 313 | 268 | 472 |
| subtotal (gross sum) | 2955 | | | | | | | | |
| total (unique sum) | 1860 | | | | | | | | |

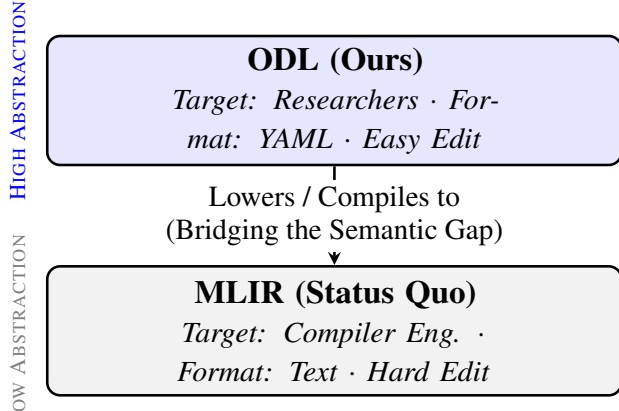

HIGH ABSTRACTION

**ODL (Ours)**
*Target: Researchers · Format: YAML · Easy Edit*

Lowers / Compiles to
(Bridging the Semantic Gap)

**MLIR (Status Quo)**
*Target: Compiler Eng. · Format: Text · Hard Edit*

LOW ABSTRACTION

*Figure 6.* System Hierarchy: ODL provides a high-level, editable YAML abstraction for Researchers, while MLIR handles the low-level execution details.

**1860 unique operators are currently mapped**; see Table 5 for a breakdown.

## 15. Extensions & Future Work

The reference implementations provided (`redacted_project_name0`, `redacted_project_name1`) operate within the Python ecosystem, but the core methodology extends far beyond it.

### 15.1. Beyond Python: Mojo and Rust

The fundamental assertion of this work is that the **ODL** is the true artifact. Because the ODL is serialized as standard YAML, writing backends for new languages is a trivial $O(1)$ engineering task. A "Rust Backend" for the engine would merely require a string template mapper similar to the existing Python one. This opens the door to transpiling high-level PyTorch research code directly into bare-metal **Mojo** or **Rust/Candle** for deployment, bypassing the Python interpreter entirely for production inference.

### 15.2. Distributed Semantic Governance

Currently, the definitions of operations like "Attention" live inside the private repositories of Google (JAX) and Meta (PyTorch). We propose a future of **Distributed Semantic Governance**, where ODL files are versioned in public Git repositories (e.g., a "HuggingFace for Semantics"). In this model, the community agrees on the mathematical definition of `FlashAttention-v3` in a central ODL repo. Framework authors then compete to provide the most efficient *implementation* of that spec, rather than competing to define the API itself. This effectively democratizes the mathematical layer of AI.

### 15.3. Model zoo porting and benchmarking at scale

Many large companies—including Google, Meta, and Hugging Face—have their own model collections ('model zoos'). For example, Hugging Face has a popular (92M downloads/month (PyPI Stats, 2026)) model zoo (Wolf et al., 2020) that Google forked and then heavily optimized for their TPU hardware and JAX (rather than PyTorch) ecosystem (Johnson & Gavrilescu, 2023).

One in-progress task of *The Authors* is to automatically port the PyTorch ecosystem optimized Hugging Face library to JAX, benchmark at scale, and then automatically port back to PyTorch. And then rinse-and-repeat for MLX, Keras, TensorFlow, &etc. Plenty of testing, sharding, and obscure edge cases remain to productionize this pipeline.

## 16. Conclusion

Futureproofing Machine Learning requires a fundamental shift in how we treat code: not as a static script, but as a mutable projection of abstract intent. We have presented a methodology centered on **Immutable Specifications**.

We have validated this methodology via the **redacted_project_name0** (Semantic) and **redacted_project_name1** (Structural) reference implementations. Whether lifting logic from dead assembly or projecting specifications via Ghost Mode, this system guarantees that the researcher's intent remains sovereign over the substrate.

All code is released open-source and without patent under the Apache-2.0 license; available at `https://github.com/` and `https://github.com/`.

## Acknowledgements

**Do not** include acknowledgements in the initial version.

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

# A. Appendix: Architecture Diagrams

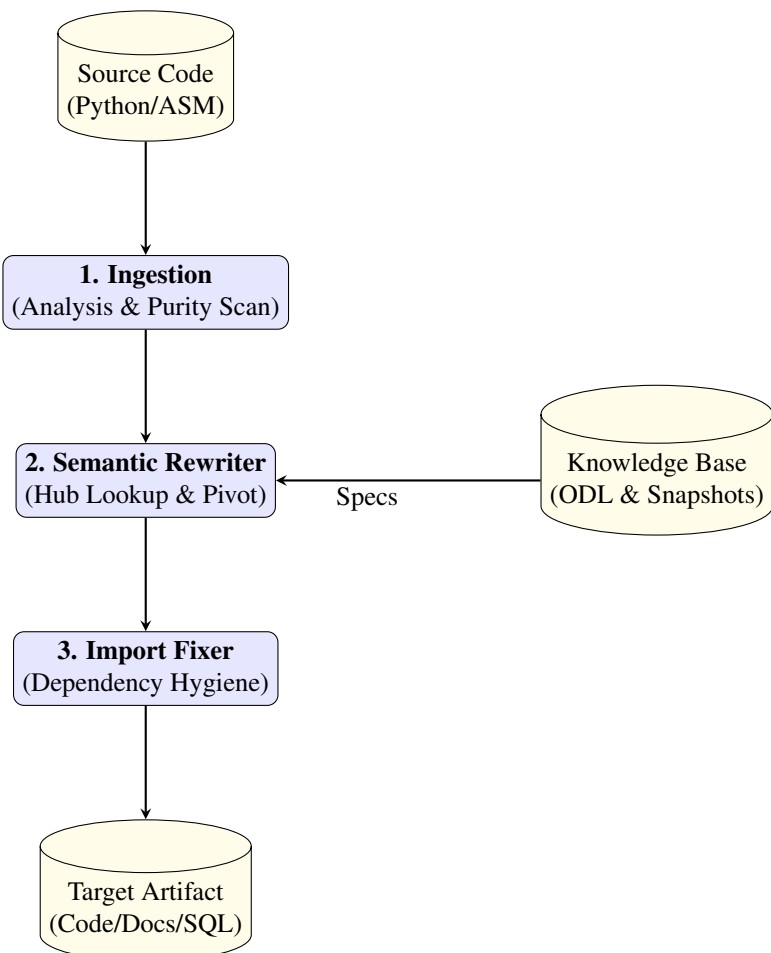

*Figure 7.* The `redacted_project_name0` Transpilation Pipeline. The central Knowledge Base drives rewriting.

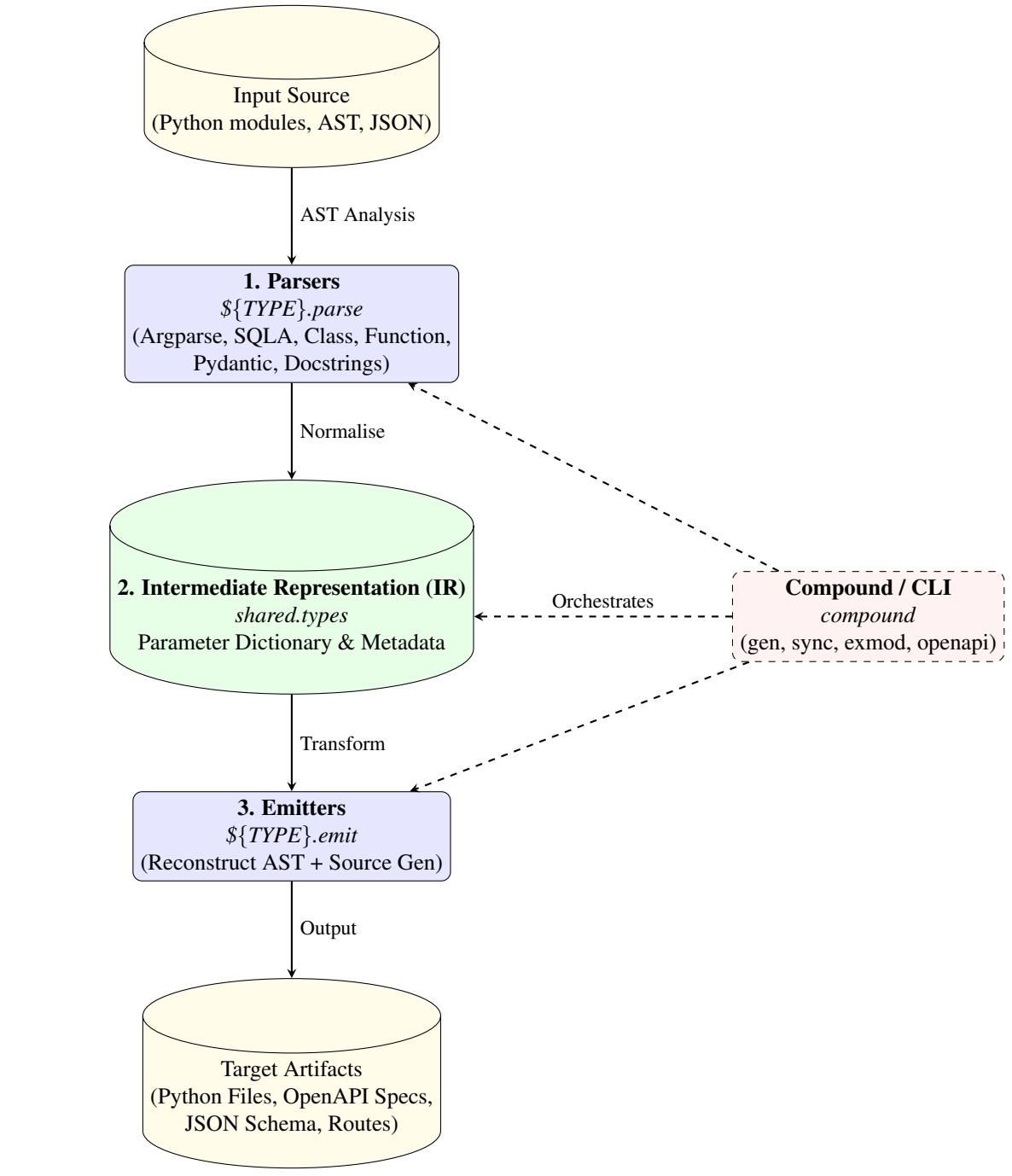

*Figure 8.* The `redacted_project_name1` Architecture. The tool parses various Python structures into a shared Intermediate Representation (IR), from which any supported output format can be emitted.

## B. Auto-Wiring Plugins

The system supports "Auto-Wiring," where plugins declare their own Semantic Specifications.

*Listing 5.* Auto-Wiring a Custom Plugin

## C. Operation Definition Language (Full Schema)

We list the full schema for reference to ensure replicability of the ODL structure.

## D. Case Study: The RNG Problem

Migrating from OOP (Torch) to Functional (JAX) requires handling Random Number Generation (RNG). Torch uses a global 'Generator' state; JAX uses stateless 'PRNGKey' passing. Our system handles this by identifying stochastic operations (like 'Dropout') in the Abstract Graph. The Rewriter Pipeline then: 1. Injects an `rng_key` argument into the 'forward' signature. 2. Injects a key splitting preamble ('k1, k2 = random.split(k)') into the function body. 3. Threads these keys into the specific calls using the ODL constraints (`requires_explicit_rng`).

## E. Case Study B: Control Flow (Imperative vs Functional)

A critical challenge in cross-framework transpilation is Control Flow. PyTorch allows imperative Python logic (`if/else`, `for` loops) that executes eagerly. JAX and TensorFlow Graph Mode require functional primitives (`jax.lax.cond`, `jax.lax.scan`) to be compiled into XLA.

This mismatch is handled by the **AuxiliaryPass** in the Rewriter Pipeline, which applies a "Static First, Safety Second" strategy.

### E.1. Static Loop Unrolling

The compiler first attempts to resolve the loop statically. If the iterator is a `range(N)` where $N$ is a literal integer constants (e.g., `for i in range(3)`), the `static_unroll` plugin activates. It unrolls the loop body $N$ times in the AST, replacing the loop variable `i` with literals $0, 1, 2$. This effectively "bakes" the control flow into a linear graph, which is valid in all target frameworks.

### E.2. Dynamic Safety and Escape Hatches

If the loop bound is dynamic (e.g., dependent on a tensor shape), automatic conversion to `jax.lax.scan` is undecidable without solving the Halting Problem (determining if/when the loop terminates).

In these cases, redacted_project_name0 refuses to guess. Instead, it utilizes the **Escape Hatch** mechanism. The imperative loop structure is preserved in the output, but wrapped in warning comments flagging it as "Runtime Side Effect." This places the burden of functionalization on the user for ambiguous cases, ensuring the compiler never produces silently incorrect logic.

### E.3. Code as Experiment Record (SQL Generation)

redacted_project_name1 projects Python configuration classes directly into **SQL Schemas**. This ensures that the experiment record layout is mathematically synchronized with the code logic of that specific commit, preventing data drift in long-running research projects.

## F. Case Study: Dataset to SQL

Consider a PyTorch 'Dataset' class representing a scientific experiment. By running 'redacted_project_name1', we transform this Logic construct into a Data storage construct (SQLAlchemy Table) without writing migration scripts.

1. **Source (Logic)**: A Python class 'ExperimentData' with typed fields in `__init__`.

2. **Extraction**: redacted_project_name1 parses the AST, identifying `self.id: int` and `self.created_at: datetime`.

3. **Projection (Data)**: The emitter generates a SQLAlchemy `Table` definition where `int` maps to `Integer` and `datetime` maps to `DateTime`.

This proves the duality of Data and Logic in the Sovereign Era: the SQL table is not a separate entity to be maintained, but a direct projection of the code structure itself.

## G. Web frontend (WebAssembly)

The hosted API docs also includes a fully embedded installation of Python linked with JavaScript and CSS to provide a fully interactive code translating and HTML rendering experience. See Figure 9.

*Figure 9.* Translation of a simple neural network from PyTorch to JAX target with its Flax NNX sub-target.

