# OpenReview forum: "Futureproofing ML: 15+ Dialects, One Compiler (PyTorch, JAX, RDNA, SASS, …)"
_ICML.cc/2026/Conference — Submitted to ICML 2026_

### Official Review · Reviewer_ntnm · 2026-02-21

**Soundness:** 2
**Presentation:** 1
**Significance:** 2
**Originality:** 2
**Overall Recommendation:** 2
**Confidence:** 4

**Summary:**

SUMMARY

Generally, this paper discusses a central concept of preserving ML research artifacts across evolving software and hardware substrates. A general topic analyzed by this study is the fragmentation of ML frameworks and the resulting "code rot" problem where research implementations become obsolete as dependencies deprecate. The authors propose a methodology based on "Sovereign Specifications" implemented through two compilers: a semantic engine supporting 15+ source/target languages spanning from high-level frameworks (PyTorch, JAX, TensorFlow) to low-level assembly (NVIDIA SASS, AMD RDNA), and a structural compiler for interface unification. The system employs a hub-and-spoke architecture with an Operation Definition Language (ODL) as the central abstraction, reducing O(N²) pairwise translation complexity to O(N). Key claimed contributions include "Ghost Mode" for framework-agnostic analysis, bidirectional translation across abstraction layers, and automated specification discovery.

**Compliance With Llm Reviewing Policy:**

Affirmed.

**Ethical Review Concerns:**

The paper lacks an ethics statement despite potential safety implications:

Automatically translated code could introduce subtle bugs in safety-critical ML applications.
Researchers may not validate translated code, trusting the tool.
Scientific results could differ due to translation errors, affecting reproducibility.
No discussion of liability or testing requirements.

**Final Justification:**

The author's final response is candid and professionally handled: they acknowledge that the remaining reviewer concerns cannot be resolved within the current submission cycle and commit to incorporating the feedback in future work. I respect this transparency.

The rebuttal did resolve the reproducibility concern: the working WASM demo and public repository confirm the system exists and partially functions, and I retracted my "project proposal" characterization accordingly. However, the three core unresolved issues from my follow-up remain open: no correctness validation table for mapped operators, no performance benchmarks for compiled code against native implementations, and no rigorous validation of assembly lifting on real production kernels. The author's acknowledgment that large-scale benchmarks against HuggingFace Transformers, MaxText, OpenVINO, and XLA are planned but not yet completed confirms that the paper is not ready for publication at this stage.

My score remains 2: Reject. This is not a judgment on the underlying engineering effort, which shows genuine ambition and practical skill, nor on the research direction, which addresses a real and underappreciated problem in ML reproducibility. The gap is strictly between the claims made and the evidence currently provided to support them. The author has a clear and actionable path to a strong systems paper: quantitative correctness evaluation, failure rate analysis via the audit subcommand, and explicit architectural differentiation from ONNX/MLIR.

I encourage resubmission to a systems or tools venue (PLDI, MLSys, or an ICML/NeurIPS workshop) once these evaluations are complete.

**Key Questions For Authors:**

Correctness Validation: How many of the 1860 mapped operators have been validated for semantic equivalence? What is your test methodology?
Real-World Success Rate: What percentage of actual research codebases (e.g., from Papers with Code) translate successfully without manual intervention?
Performance: What is the compilation time overhead? How does generated code performance compare to native implementations?
Assembly Lifting: Can you provide validation of SASS/RDNA lifting on real production kernels (e.g., cuDNN, rocBLAS)? The toy examples are unconvincing.
Maintenance: Who maintains the 15+ framework snapshots? How do you handle API changes? What is the update cadence?
MLIR Integration: Why create a competing IR instead of contributing high-level dialects to MLIR? Could ODL compile to MLIR?
Failure Analysis: What are the most common translation failures? What percentage of operations have no ODL mapping?
Numerical Precision: How do you handle framework-specific numerical behaviors (e.g., PyTorch's default float32 vs JAX's bfloat16)?
Backward Compatibility: Have you successfully run 2018-era code in 2026 environments? Can you provide concrete examples?
Self-Healing Concerns: How do automatically relaxed tolerances (Section 12) avoid introducing silent numerical errors?

**Limitations:**

Add Comprehensive Evaluation:
Correctness validation on benchmark suite (100+ real models). Performance benchmarks (compilation time, runtime overhead). Case studies showing successful translation of published research code. User study with independent researchers. Failure rate analysis


Validate Assembly Lifting: The claim of lifting SASS/RDNA is extraordinary and requires extraordinary evidence. Either provide rigorous validation or remove this claim.
Compare with Baselines: Systematic comparison with ONNX, MLIR, framework-native export tools, and manual porting effort.
Provide Correctness Guarantees: Formal or empirical evidence that translations preserve semantics. What are the limitations?
Address Maintenance: Realistic assessment of engineering burden to maintain 15+ targets. Who will do this work?
Simplify Presentation: Remove grandiose language, focus on technical precision. The "Sovereign Era" framing is distracting.
Add Ethics Statement: Discuss implications of automated translation potentially introducing bugs in scientific code.
Fix Technical Gaps: Complete algorithm specifications, provide implementation details, explain failure modes.

Recommended Venue:
This work might be better suited for: Systems venues (OSDI, SOSP, ATC) with focus on implementation and performance; Tools venues (PLDI, CGO) with emphasis on compilation techniques; Workshops (MLSys workshop at NeurIPS/ICML) for initial feedback before full conference submission.

ADDITIONAL COMMENTS

The fundamental issue is a mismatch between ambition and validation. The paper proposes an extremely ambitious system (15+ targets, assembly to high-level, futureproofing) but provides almost no evidence it works beyond operation counts.

The strongest version of this paper would:
Narrow scope to high-level framework translation (PyTorch ↔ JAX ↔ TensorFlow). Provide rigorous correctness evaluation. Show performance is acceptable. Demonstrate real-world utility with case studies. Integrate with (or clearly differentiate from) MLIR/ONNX.

The current version reads more like a vision paper or project proposal than a validated research contribution. With 12+ months of rigorous evaluation and significant scope reduction, this could become a solid systems paper.

**Strengths And Weaknesses:**

EVALUATION STRENGTHS

Important Problem: The paper addresses a genuine challenge in ML research - the rapid obsolescence of implementation code despite stable underlying mathematics. This is a practically relevant concern for reproducibility and long-term research value.

Scope: Supporting 15+ frameworks/targets across multiple abstraction levels (from PyTorch to SASS assembly) demonstrates significant engineering effort and breadth.

Novel Architectural Contributions:
1. The hub-and-spoke ODL approach is cleaner than N^2 pairwise converters.
2. "Ghost Mode" using serialized snapshots instead of runtime introspection is a creative approach.
3. Bidirectional assembly lifting (SASS/RDNA to high-level specs) is interesting if validated.

Practical Implementation: The authors promise open-source release under Apache 2.0, which could benefit the community if the system works as claimed.
System Design: The integration of semantic compilation, structural regularization, and automated discovery shows thoughtful architecture.
Motivation: Section 2's history of ML frameworks effectively motivates the problem, and Section 3's "Code Rot Paradox" provides good theoretical framing.

MAJOR WEAKNESSES

1. Inadequate Empirical Evaluation: This is the paper's most critical flaw. For a systems paper claiming to solve code translation across 15+ targets, the evaluation is shockingly minimal:

a) No Correctness Validation: Table 5 shows 1860 operators mapped, but provides zero evidence these translations are correct. How many produce semantically equivalent code? What is the error rate? Where is this table? Moreover, paper doesn't have some results and visualizations such as Tables 1-4 (becase we have table 5), Figs. 3, 5, 9, listings 1-4.

b) No Performance Benchmarks: What is compilation time? (critical for practical use). What is runtime overhead of generated code? How does translated code compare to hand-written native code?

c) No Real-World Case Studies: The paper lacks end-to-end examples of actual research code being successfully translated and executed. Claims about "futureproofing" are unsubstantiated without showing 2018 code working in 2026.

d) No User Studies: Has anyone besides the authors used this system? What is the learning curve? What percentage of real codebases translate successfully?

e) No Baseline Comparisons: How does this compare to ONNX for model exchange? To manual porting effort? To framework-specific tools like PyTorch's export utilities?

2. Unvalidated Core Claims. Several bold claims lack supporting evidence:

a) "Digital Archaeology" (Assembly Lifting): Section 9 claims to lift NVIDIA SASS and AMD RDNA back to high-level logic. This is extraordinarily difficult. The examples in Listing 3/4 and Figure 3 (which are absent) are trivial toys. Where is validation on real kernels? What percentage of actual compiled code can be lifted? This claim borders on implausible without substantial evidence.

b) Semantic Equivalence: The paper assumes ODL captures mathematical intent, but provides no formal verification. How do you guarantee that torch.matmul and jax.numpy.matmul have identical semantics given different broadcasting rules, dtype handling, and numerical precision?

c) "Ghost Mode" Reliability: Claims of analyzing code "without that framework being executed or even installed" depend entirely on snapshot completeness. What happens when APIs have runtime-dependent behavior? When do snapshots become stale?

3. Missing Technical Details. Critical implementation information is absent:

ODL Expressiveness: What constructs cannot be represented? The paper mentions control flow challenges (Appendix E) but doesn't quantify limitations.
Failure Modes: What percentage of real code translates successfully? The "Escape Hatch" mechanism (Appendix E.2) suggests frequent failures, but no statistics are provided.
Correctness Guarantees: The "Self-Healing Specs" (Section 12) that automatically relax tolerances when tests fail is alarming. This could silently introduce numerical errors.
Snapshot Maintenance: Who creates/maintains the 15+ framework snapshots? How often do they need updating? This seems like a massive ongoing engineering burden.

4. Comparison with Existing Solutions is Superficial. Section 13 dismisses MLIR in two paragraphs, claiming it's "designed for Compiler Engineers" while ODL targets "Researchers." This is a false dichotomy:
MLIR has high-level dialects (Torch-MLIR, MHLO) specifically for researchers
ONNX already solves model exchange between frameworks
StableHLO provides framework-agnostic representations
Each framework has export mechanisms (torch.export, jax.jit)
The paper doesn't explain why a new system is needed instead of contributing to these established efforts. The comparison feels dismissive rather than analytical.

5. Grandiose Writing Obscures Substance

"The Methodology of 'Done'" (Section 1 title) is vague
"Declarative Sovereignty," "Sovereign Era," "Digital Archaeology" - buzzwords without precise definitions.
"Time-Travelling Sovereign Era" (Section 2.4) is marketing, not science. The writing would benefit from focusing on technical precision over rhetorical flourish.

6. Questionable Design Decisions

YAML for ODL: Section 14 justifies YAML for "Human-Readability and Fuzzability," but YAML is notoriously problematic (the "Norway Problem," security issues). Why not use established schema languages like Protocol Buffers or JSON Schema?
Plugin Architecture Complexity: The extensive plugin system (RNG threading, purity analysis, graph optimization) suggests the core translation is insufficient and requires many special cases.
"Self-Healing" Specs: Automatically relaxing numerical tolerances when tests fail (Section 12) is dangerous and could mask bugs.

7. Scalability and Maintenance Concerns

Supporting 15+ targets requires maintaining 15+ code generators, parsers, and snapshots. Framework APIs change constantly (PyTorch 2.0 → 2.1 → 2.2). No discussion of how the project would handle this maintenance burden. The claim that adding new languages is "trivial O(1) engineering" (Section 15.1) is naive.

8. Missing Related Work. The paper ignores significant relevant work:

Cross-compilation: ONNX Runtime, TensorRT, OpenVINO.
Intermediate Representations: MLIR in depth, TVM, Glow, XLA/HLO.
Program Synthesis: Sketch, Rosette, neural program synthesis.
Static Analysis: Abstract interpretation, symbolic execution for ML.
Legacy Code Migration: Software modernization literature.

9. Reproducibility Concerns. Despite promises of open-source release:

GitHub links are redacted (understandable for blind review but concerning). No supplementary materials provided in sufficient quantityю. No dataset of test cases. Listing 1, 2, 5, 6 show partial code snippets, not complete implementations. Figure 5's table shows counts but no test suite or validation data.

10. Ethical and Practical Implications Unexamined

What happens when automatically translated code has subtle bugs?
Who is liable if research results differ due to translation errors?
Could this enable academic fraud (running experiments in one framework, claiming another)?
No discussion of testing/validation requirements for translated code.

MINOR WEAKNESSES

Notation Inconsistencies: Switches between mathematical notation (G for graphs) and code snippets without clear transitions.
Figure Quality: Figure 5's table is hard to read with small font; Figure 9 (Screenshot) doesn't clearly demonstrate functionality.

Algorithm Specifications: Algorithm 1 (Semantic Pivot) is high-level pseudocode, lacks detail. Algorithm 2 (Graph Fusion) is trivial. Algorithm 3 (Structural Projection) is incomplete.

Appendix Organization: Case studies in appendices feel like afterthoughts rather than validated examples. Citation Issues: Some citations are incomplete or informal (e.g., "PyPi, S." as author).

TECHNICAL SOUNDNESS

Conceptual Issues: Semantic Equivalence Assumption: The core premise that mathematical operations have universal definitions is flawed. Different frameworks make different trade-offs: numerical precision (float32 vs bfloat16 defaults), broadcasting semantics
Gradient computation rules, memory layouts (NHWC vs NCHW).

Control Flow Translation: The paper acknowledges this is undecidable (Appendix E.2) but doesn't quantify how often this occurs in practice.
State Management: OOP vs Functional paradigm mismatch (PyTorch vs JAX) requires fundamental restructuring, not just API mapping. The RNG example (Appendix D) is one case, but what about optimizer states, batch normalization running statistics, etc.?
Assembly Lifting: Claiming to reverse-engineer high-level intent from optimized assembly (Section 9) is extremely ambitious. Compiler optimizations (loop unrolling, vectorization, fusion) destroy high-level structure. The trivial examples provided don't validate this capability.

COMPLIANCE WITH ICML AUTHOR INSTRUCTIONS

Format: Generally compliant with ICML LaTeX template (8 pages + references + appendix).
Anonymization: Properly anonymized with redacted project names.
Page Limit: Appears to comply (8 pages main content).
Ethics Statement: Missing. Given potential for introducing subtle bugs in critical ML systems, ethical considerations should be discussed.
Reproducibility: Partially compliant. Algorithms provided but incomplete. Code promised but not available for review.

Violations:
Missing ethics statement for a systems paper with potential safety implications.

COMPARISON TO CURRENT RESEARCH STANDARDS

ML Compilers and IRs: The field has converged on: MLIR as the standard multi-level IR framework; ONNX for model exchange; Framework-specific export mechanisms (torch.export, saved_model); Specialized compilers (TVM, XLA, TensorRT).

This paper's ODL approach positions itself as complementary but doesn't integrate with these standards or explain why incompatibility is acceptable. Existed Software Translation Research in:

Source-to-source compilation (C2Rust, Java2Kotlin). API migration (Android API updates, library modernization). Program synthesis and sketch-based tools.
Typically emphasizes correctness guarantees and extensive validation. This paper provides neither.

Reproducibility Crisis in ML community is increasingly focused on:

Containerization (Docker, Singularity). Dependency pinning (conda, poetry). Execution environments (Papers with Code, CodaLab).
A translation-based approach is orthogonal and unproven compared to these established practices.

---

> ### Author Rebuttal · Authors · 2026-03-24
>
> I have requested special dispensation from the ICML committee to share these unredacted links with you. Because of the reproducibility problem you mention as your reason for rejection.
>
> Here is a live WASM (WebAssembly) demo: https://samuelmarks.github.io/ml-switcheroo/
>
> Here is the code: https://github.com/SamuelMarks/ml-switcheroo
>
> Yes, code is always a projection of some abstract intent; but here the goal is to abstract away the differences between frameworks so the algorithm is translates rather than some pecularity of the framework (like data types, rng threading, functional vs class based OOP, fusion operations rather than many smaller operations, etc.).
>
> NVIDIA for example released Tile for Python a little while ago, and a few days ago released the same for C++. This is a whole new programming paradigm for CUDA. ml-switcheroo could be extended to translate to/fro that new programming paradigm. That's the unique value proposition of ml-switcheroo; no longer are we tied into some specific programming paradigm, framework version, or even programming language. With ml-switcheroo all you need is a lossless intermediate representation that is maintained, works in ghost-mode (no frameworks installed; e.g., in the browser with wasm), and (maybe) writing a few plugins that cleanly hook into the existing library structure.
>
> Microsoft just interviewed me for a Principal Engineer in their ONNX team. I built onnx9000 for it. In the near future I may fold ml-switcheroo into it; or it into ml-switcheroo. onnx9000 currently doesn't have a strong focus on precise API compatibility like ml-switcheroo does [ml-switcheroo even has an `audit` subcommand to analyse a repo for API calls unmapped my ml-switcheroo].
> On the otherhand, onnx9000 is more of a proof-of-concept for using ONNX as intermediate representation in the browser; for framework translation, model optimisation and analysis, training, and inference. Though it does some bidirectional compilation/transpilation, this is not the focus; and may be lossy. https://github.com/SamuelMarks/onnx9000
>
> Your feedback has weight, and I can tone down the rhetoric throughout and focus more on the escape-hatch cases; I.e., where ml-switcheroo fails.
>
> ml-switcheroo is not a project proposal, like you suggested. It works. Is it API complete? - Well… almost. - As I reference in future work, it is almost capable of taking a big model zoo (like https://github.com/huggingface/transformers ) and translating it into a model zoo for a different framework (e.g., Google ecosystem JAX Flax NNX ). And at scale. This is related to my previous commercial work last year at Google on their https://github.com/AI-Hypercomputer/maxtext project.
>
> I have requested special dispensation from the ICML committee to share these unredacted links with you.
>
> With your permission, I will modify my submission such that it addresses most all of your concerns.

---

> > ### Author Rebuttal · Reviewer_ntnm · 2026-04-05
> >
> > Thank you for the rebuttal and for obtaining dispensation to share the links. I have reviewed both the live WASM demo (https://samuelmarks.github.io/ml-switcheroo/) and the source repository (https://github.com/SamuelMarks/ml-switcheroo).
> >
> > What is now resolved:
> > The existence and reproducibility concern is substantially mitigated. The working WASM demo is a concrete, verifiable artifact. The repository structure is coherent and aligns with the architectural claims in the paper. I retract my concern that this reads as a "project proposal"; the system demonstrably exists.
> >
> > What remains unresolved:
> > Correctness validation is still absent. The repository and demo do not include a benchmark suite demonstrating that the 1,860 mapped operators produce semantically equivalent outputs across source/target pairs. Showing that translation runs is not equivalent to showing it is correct. A table of pass/fail rates on a held-out operator test suite is the minimum required for a systems paper.
> >
> > Performance data is still absent. Compilation latency and runtime overhead of generated code relative to native implementations are not provided either in the rebuttal or the repository.
> >
> > Assembly lifting (Section 9) remains unvalidated. No evidence is provided that SASS/RDNA lifting works on real production kernels beyond the toy examples in Listings 3-4.
> >
> > The MLIR/ONNX comparison is still superficial. The mention of onnx9000 in the rebuttal actually strengthens the concern: if the author is separately building an ONNX-based tool for browser inference, the boundary between these systems and the claimed differentiation from ONNX in Section 13 needs to be made precise in the paper.
> >
> > Follow-up questions for the authors:
> >
> > Can you provide a table of correctness test results (not just operator counts) for at minimum the PyTorch → JAX translation path, tested against real models from a public benchmark (e.g., a subset of HuggingFace transformers)? What is the documented failure rate of ml-switcheroo on real codebases? The audit subcommand you mention is exactly the right tool for this - please include its output on a representative codebase in the revision. How does ml-switcheroo relate to onnx9000? If there is architectural overlap, this must be explicitly stated in the related work section.
> >
> > I am open to revising my score upward if the revision addresses points 1 and 2 with quantitative data.

---

> > > ### Author Response · Authors · 2026-04-05
> > >
> > > Thanks for your response. You are a very good reviewer and I want to respond to each in turn. Unfortunately the other reviewers are not budging, so I don't see a path forward to getting this paper accepted into this ICML.
> > >
> > > In the meantime, I'll continue working on onnx9000 and ml-switcheroo and potentially combine the two into one mega project. Next time I submit—whether it be to ICML or elsewhere—I'll incorporate your feedback. In particular this should give me time to run large scale benchmarks and comparisons with other model zoos (huggingface transformers, maxtext, etc.) and other compilers (openvino, xla, etc.) &etc.

---

### Official Review · Reviewer_wMYb · 2026-03-10

**Soundness:** 1
**Presentation:** 1
**Significance:** 2
**Originality:** 2
**Overall Recommendation:** 1
**Confidence:** 4

**Summary:**

The paper motivates a method to “Futureproof Machine Learning” by lifting logic from programming languages into semantic (stage 1) and structure (stage 2) intermediate representations. These intermediate representations are structured as YAML documents rather than another executable language itself. The document, absent of the need to be interpretable or parseable, is persistent across time as an artifact of the software intent.

**Compliance With Llm Reviewing Policy:**

Affirmed.

**Final Justification:**

The rebuttal provided an updated link for the missing code, which resolves some of my concerns about the reproducibility and legitimacy of the claims made. However, most of my initial concerns remain. Most concerning to me is that evaluations are sparse and unclear in the paper. I recommend the author keep up the ambition and engineering work, but take a look at prior published papers to see what is demanded to submit it as a research artifact.

**Key Questions For Authors:**

* Figure 6 claims that text is harder to edit than yaml. Why is that true?
* What is the key difference between the IRs in the two stages?
* Why are we redacting the names of the components?

**Limitations:**

No. Consider how well a high-level specification would survive time as well. Absent of grounded semantics, how would such a document impact the responsibilities of software developers?

**Strengths And Weaknesses:**

## Strengths
The motivation is clear and several concrete examples for its necessity are provided.


## Weaknesses
* The entire structure and sentence-level style of the text feels AI-written.
* Many elements are missing from the document. For example, all Listings are only the caption without any accompanying content. Figure 3 has no accompanying figure. Figure 9 has only the ghost formatting figure. All citations seem to be only text, without the hyperlink that usually comes with `~\cite{}` to the bibtex entry.
* The conclusion shares the “open-sourced code” to a link that is simply the link to github.com
* The motivation of a single “lifted” representation is not new. Verified lifting has been studied for many years, yet a proper citation or reference to it in the background is missing from the paper.
* No rubric or description of evaluation. How is “successfully mapped” evaluated?

---

> ### Author Rebuttal · Authors · 2026-03-24
>
> I have requested special dispensation from the ICML committee to share these unredacted links with you. Because of the reproducibility problem you mention as your reason for rejection. A lot of the figures and such were redacted because they included github links and the like; enabling deanonymisation of the paper.
>
> Here is a live WASM (WebAssembly) demo: https://samuelmarks.github.io/ml-switcheroo/
>
> Here is the code: https://github.com/SamuelMarks/ml-switcheroo
>
> Yes, code is always a projection of some abstract intent; but here the goal is to abstract away the differences between frameworks so the algorithm is translates rather than some pecularity of the framework (like data types, rng threading, functional vs class based OOP, fusion operations rather than many smaller operations, etc.).
>
> NVIDIA for example released Tile for Python a little while ago, and a few days ago released the same for C++. This is a whole new programming paradigm for CUDA. ml-switcheroo could be extended to translate to/fro that new programming paradigm. That's the unique value proposition of ml-switcheroo; no longer are we tied into some specific programming paradigm, framework version, or even programming language. With ml-switcheroo all you need is a lossless intermediate representation that is maintained, works in ghost-mode (no frameworks installed; e.g., in the browser with wasm), and (maybe) writing a few plugins that cleanly hook into the existing library structure.
>
> Microsoft just interviewed me for a Principal Engineer in their ONNX team. I built onnx9000 for it. In the near future I may fold ml-switcheroo into it; or it into ml-switcheroo. onnx9000 currently doesn't have a strong focus on precise API compatibility like ml-switcheroo does [ml-switcheroo even has an `audit` subcommand to analyse a repo for API calls unmapped my ml-switcheroo].
> On the otherhand, onnx9000 is more of a proof-of-concept for using ONNX as intermediate representation in the browser; for framework translation, model optimisation and analysis, training, and inference. Though it does some bidirectional compilation/transpilation, this is not the focus; and may be lossy. https://github.com/SamuelMarks/onnx9000
>
> Your feedback has weight, and I can tone down the rhetoric throughout and focus more on the escape-hatch cases; I.e., where ml-switcheroo fails. And I can expand on the RDNA to PyTorch, and SASS to PyTorch scenarios also. ("PyTorch" can be replaced with any other framework in ml-switcheroo)
>
> ml-switcheroo is not a project proposal. It works. Is it API complete? - Well… almost. - As I reference in future work, it is almost capable of taking a big model zoo (like https://github.com/huggingface/transformers ) and translating it into a model zoo for a different framework (e.g., Google ecosystem JAX Flax NNX ). And at scale. This is related to my previous commercial work last year at Google on their https://github.com/AI-Hypercomputer/maxtext project.
>
> I have requested special dispensation from the ICML committee to share these unredacted links with you.
>
> With your permission, I will modify my submission such that it addresses most all of your concerns.

---

> > ### Author Rebuttal · Reviewer_wMYb · 2026-04-03
> >
> > Thanks for the additional context and code link. However, the remainder of the concerns in my original review are still unresolved so I maintain my score.

---

### Official Review · Reviewer_FJwx · 2026-03-11

**Soundness:** 1
**Presentation:** 2
**Significance:** 1
**Originality:** 3
**Overall Recommendation:** 1
**Confidence:** 5

**Summary:**

This paper proposes a metalanguage for unified ML frameworks and describes a compiler from common frameworks to this metalanguage. The authors claim that their language can act as a universal framework for ML specifications that can be used to future-proof them as frameworks and underlying hardware change over time.

**Compliance With Llm Reviewing Policy:**

Affirmed.

**Key Questions For Authors:**

Why privilege your new proposed format as the ideal metalanguage for mapping between formats? Why not do that with some existing format? (e.g., Section 13 is vague and unconvincing)

What testable claims can you make in favor of your method? What results could you then provide to test these claims?

> the utility of ML source code decays at a significantly accelerated rate compared to traditional systems software

What evidence do you have for this?

> the optimal expression of logic changes, rendering previous syntactic abstractions obsolete or inefficient

While Nvidia hardware has changed significantly in the last few years, Pytorch syntax has remained largely stable. This is a strong claim that I don't immediately accept, and which you should either spend more time justifying or revise

> The fundamental assertion of this work is that the ODL is the true artifact. Because the ODL is serialized as standard YAML, writing backends for new languages is a trivial O(1) engineering task.

The notion of an O(1) engineering task is nice precise

> Futureproofing Machine Learning requires a fundamental shift in how we treat code: not as a static script, but as a mutable projection of abstract intent.

Code is always a projection of some abstract intent?

**Limitations:**

yes

**Strengths And Weaknesses:**

## Strengths

The idea proposed by the authors is interesting and worth exploring further.

## Weaknesses

I am not convinced that the framework proposed here should be privileged as universal relative to other frameworks that have been proposed in the past. Any framework can claim that they should be privileged as a metalanguage that other approaches should be mapped into. I don't see a clear reason why this language should be somehow unique. Ideally, the paper would clearly explain what properties this metalanguage satisfies, either with clear comparison to prior approaches or experiments that validate testable claims. In the current draft, both the description of the metalanguage are vague and its design principles are quite general. e.g., any software library presumably attempts to ensure two of the three key properties: structural regularity ("Interfaces must be consistent") and semantic sovereignty ("the abstract definition of an operation is sovereign over its implementation"). The third desideratum of representational independence ("Mathematical logic must be stored in a format that does not require a Turing-complete interpreter to parse") is more specific, but the assertion that not being Turing-complete will make software more future proof is not sufficiently justified.

I've left some questions and comments below under "Key Questions" as constructive feedback towards developing the motivation and justification of the project.

---

> ### Author Rebuttal · Authors · 2026-03-24
>
> I have requested special dispensation from the ICML committee to share these unredacted links with you. Because of the reproducibility problem you mention as your reason for rejection.
>
> Here is a live WASM (WebAssembly) demo: https://samuelmarks.github.io/ml-switcheroo/
>
> Here is the code: https://github.com/SamuelMarks/ml-switcheroo
>
> Yes, code is always a projection of some abstract intent; but here the goal is to abstract away the differences between frameworks so the algorithm is translates rather than some pecularity of the framework (like data types, rng threading, functional vs class based OOP, fusion operations rather than many smaller operations, etc.).
>
> NVIDIA for example released Tile for Python a little while ago, and a few days ago released the same for C++. This is a whole new programming paradigm for CUDA. ml-switcheroo could be extended to translate to/fro that new programming paradigm. That's the unique value proposition of ml-switcheroo; no longer are we tied into some specific programming paradigm, framework version, or even programming language. With ml-switcheroo all you need is a lossless intermediate representation that is maintained, works in ghost-mode (no frameworks installed; e.g., in the browser with wasm), and (maybe) writing a few plugins that cleanly hook into the existing library structure.
>
> Microsoft just interviewed me for a Principal Engineer in their ONNX team. I built onnx9000 for it. In the near future I may fold ml-switcheroo into it; or it into ml-switcheroo. onnx9000 currently doesn't have a strong focus on precise API compatibility like ml-switcheroo does [ml-switcheroo even has an `audit` subcommand to analyse a repo for API calls unmapped my ml-switcheroo].
> On the otherhand, onnx9000 is more of a proof-of-concept for using ONNX as intermediate representation in the browser; for framework translation, model optimisation and analysis, training, and inference. Though it does some bidirectional compilation/transpilation, this is not the focus; and may be lossy. https://github.com/SamuelMarks/onnx9000
>
> Your feedback has weight, and I can tone down the rhetoric throughout and focus more on the escape-hatch cases; I.e., where ml-switcheroo fails.
>
> I have requested special dispensation from the ICML committee to share these unredacted links with you.
>
> With your permission, I will modify my submission such that it addresses most all of your concerns.

---

> > ### Author Rebuttal · Reviewer_FJwx · 2026-03-31
> >
> > Thanks for the comments. I stand by my current assessment of the manuscript for the purposes of ICML and encourage future revisions based on the feedback from the reviewers.

---

### Official Review · Reviewer_XUqk · 2026-03-14

**Soundness:** 2
**Presentation:** 1
**Significance:** 2
**Originality:** 4
**Overall Recommendation:** 2
**Confidence:** 3

**Summary:**

The paper is essentially a design document for a new abstract model for machine learning frameworks. It argues that logic should be extracted into an ODL format, which is then synthesized into lower-level code components.

**Compliance With Llm Reviewing Policy:**

Affirmed.

**Final Justification:**

The rebuttal mainly provided the code for the concept that was proposed. The authors have not addressed my questions about robustness and adapting to AI-generated code, nor has it provided a full implementation of all of the components of the pipeline (only a few simple ones, as the demo).

**Key Questions For Authors:**

- How do you deal with the fact that as libraries build up, new abstractions will be required to represent certain constructs?
- We are moving into an era where AI will write more and more of the world's code. How will your framework remain robust to this shift, and how will you ensure issues in various parts of the pipeline be fixed as quickly and effortlessly as possible?

**Limitations:**

no limitations section

**Strengths And Weaknesses:**

Soundness:
- A core claim is reducing the O(N^2) complexity to the O(N) hub and spoke model. While this seems nice in theory, there is no evidence for a good unified representation of the proposed abstract logic hub that will easily translate to the representations. In addition, in this model, translating logic to representations seems more complex than in the O(N^2) model.

Presentation:
- The ideas in the paper is presented at a very high level, with very few concrete examples and details. Some aspects of the paper are difficult to follow.
- I do not think I can reproduce any aspect of the proposed framework due to the lack of details on concrete implementation.
- This paper might be better suited for a conference's position paper track.

Significance:
- Without any concrete proof of concept, the significance of the proposed ideas are unclear.

Originality:
- I find the ideas in this work to be very original, proposing a new way of thinking about ML frameworks and code maintainability moving forward.

---

> ### Author Rebuttal · Authors · 2026-03-24
>
> I have requested special dispensation from the ICML committee to share these unredacted links with you. Because of the reproducibility problem you mention as your reason for rejection.
>
> Here is a live WASM (WebAssembly) demo: https://samuelmarks.github.io/ml-switcheroo/
>
> Here is the code: https://github.com/SamuelMarks/ml-switcheroo
>
> Microsoft just interviewed me for a Principal Engineer in their ONNX team. I built onnx9000 for it. In the near future I may fold ml-switcheroo into it; or it into ml-switcheroo. onnx9000 currently doesn't have a strong focus on precise API compatibility like ml-switcheroo does [ml-switcheroo even has an `audit` subcommand to analyse a repo for API calls unmapped my ml-switcheroo].
> On the otherhand, onnx9000 is more of a proof-of-concept for using ONNX as intermediate representation in the browser; for framework translation, model optimisation and analysis, training, and inference. Though it does some bidirectional compilation/transpilation, this is not the focus; and may be lossy.
>
> Your feedback has weight, and I can tone down the rhetoric throughout and focus more on the escape-hatch cases; I.e., where ml-switcheroo fails.
>
> I have requested special dispensation from the ICML committee to share these unredacted links with you.
>
> With your permission, I will modify my submission such that it addresses most all of your concerns.

---

> > ### Author Rebuttal · Reviewer_XUqk · 2026-04-03
> >
> > Despite the authors sharing their implementation, I did not see any obvious comparison or benefit of this model, as I mentioned in the review. I believe showing the model's superiority requires detailed comparisons with related or similar baselines and systems.

---

### Decision · Program_Chairs · 2026-04-30

**Decision:**

Reject

**Comment:**

While reviewers agreed that this paper tackle an interesting problem, and that the paper contains interesting ideas which have a good potential, they also raised serious concerns about the paper's significance, soundness, and presentation. In particular, reviewers are concerned about the motivation for adoption of the proposed framework, the limited evaluation of the proposed framework, and about the presentation which misses important details. Therefore, the paper cannot be accepted to the conference at this time.